

# Democratizing AI: non-expert design of prediction tasks

James P. Bagrow[1,2]

[1] Mathematics & Statistics, University of Vermont, Burlington, VT, USA
[2] Vermont Complex Systems Center, University of Vermont, Burlington, VT, USA

## ABSTRACT

Non-experts have long made important contributions to machine learning (ML) by contributing training data, and recent work has shown that non-experts can also help with feature engineering by suggesting novel predictive features. However, non-experts have only contributed features to prediction tasks already posed by experienced ML practitioners. Here we study how non-experts can design prediction tasks themselves, what types of tasks non-experts will design, and whether predictive models can be automatically trained on data sourced for their tasks. We use a crowdsourcing platform where non-experts design predictive tasks that are then categorized and ranked by the crowd. Crowdsourced data are collected for top-ranked tasks and predictive models are then trained and evaluated automatically using those data. We show that individuals without ML experience can collectively construct useful datasets and that predictive models can be learned on these datasets, but challenges remain. The prediction tasks designed by non-experts covered a broad range of domains, from politics and current events to health behavior, demographics, and more. Proper instructions are crucial for non-experts, so we also conducted a randomized trial to understand how different instructions may influence the types of prediction tasks being proposed. In general, understanding better how non-experts can contribute to ML can further leverage advances in Automatic machine learning and has important implications as ML continues to drive workplace automation.

Corresponding author
James P. Bagrow,
james.bagrow@uvm.edu

## INTRODUCTION

Recent years have seen improved technologies geared towards workplace automation and there is both promise and peril in how AI, robotics, and other technologies can alter the job security and prospects of the future workforce (*David, 2015*; *Frank et al., 2019*). While automation has already upended factory work and other traditionally blue collar jobs, machine learning (ML) can have similar effects on offices and knowledge work. ML can enable firms to better understand and act on their data more quickly and perhaps with fewer employes. But how well can or should employes understand the process and scope of ML? Perhaps most importantly, on their own, can individuals apply ML in new problem

areas, informed by their own domain knowledge, or is such "editorial" control of ML limited to experts with significant training and experience in ML and related areas?

At the same time, ML itself is being automated (*Feurer et al., 2015*), and the emerging field of Automatic machine learning (AutoML) promises to lower the barrier of entry at least to some extent, and in time the role of ML expertise may be supplanted by sufficiently advanced "AutoML" methods. Yet non-experts (meaning those with little prior experience in ML) will not be able to apply ML to a new area of their own interest if they can only contribute features to pre-existing problems in pre-existing areas. To fully leverage advances in ML, non-experts must be able to define their own tasks in new areas.

In this article, we investigate the following research questions (RQs):

1. Can individuals who are not experts in the details of statistical or ML design meaningful supervised learning prediction tasks?
2. What are the properties of prediction tasks proposed by non-experts? Do tasks tend to have common properties or focus on particular topics or types of questions?
3. Does providing an example of a prediction task help clarify the design assignment for non-experts or do such examples introduce bias?
4. Are non-experts able to compare and contrast new prediction tasks, to determine which tasks should be deemed important or interesting?
5. Can data collected for proposed tasks be used to build accurate predictive models without intervention from an ML expert?

To study these questions, we employ a crowdsourcing platform where groups of crowd workers ideate supervised learning prediction tasks (RQ1), categorize and efficiently rank those tasks according to criteria of interest (RQ4), and contribute training data for those tasks. We study the topics and properties of prediction tasks (RQ2), show that performant predictive models can be trained on some proposed tasks (RQ5), and explore the design of the problem of proposing prediction tasks using a randomized trial (RQ3). We discuss limitations and benefits when approaching learning from this perspective—how it elevates the importance of specifying prediction task requirements relative to feature engineering and modeling that are traditionally the focus of applied ML.

## BACKGROUND

Machine learning requires experts to understand technical concepts from probability and statistics, linear algebra and optimization; be able to perform predictive model construction, training, and diagnostics; and even participate in data collection, cleaning, and validation (*Domingos, 2012*; *Alpaydin, 2020*). Such a depth of pre-requisite knowledge may limit the roles of non-experts, yet fields such as AutoML (*Hutter, Kotthoff & Vanschoren, 2019*) have the promise to further enable non-expert participation in ML by removing many of the challenges which non-experts may be unable to address without training or experience (*Feurer et al., 2015*; *Vanschoren et al., 2014*). Indeed, understanding the role of non-experts in ML is increasingly important as ML become more ubiquitous and affects the future of work (*Frank et al., 2019*). Non-experts have long been able to

participate in data collection to train predictive models and interactive ML allows non-experts and ML to work together to better address pre-identified problems (*Fails & Olsen, 2003*; *Cheng & Bernstein, 2015*; *Crandall et al., 2018*). However, as remarked by *Yang et al. (2018)*, despite extensive research in these areas, little work has investigated how non-experts can take creative or editorial control to design their own applications of ML.

Crowdsourcing has long been used as an avenue to gather training data for ML methods (*Lease, 2011*). In this setting, it is important to understand the quality of worker responses, to prevent gathering bad data and to maximize the wisdom-of-the-crowd effects without introducing bias (*Hsueh, Melville & Sindhwani, 2009*). Researchers have also studied active learning in this context, where an ML method is coupled in a feedback loop with responses from the crowd. One example is the Galaxy Zoo project (*Kamar, Hacker & Horvitz, 2012*), where crowd workers are asked to classify photographs of galaxies while learning algorithms try to predict the galaxy classes and also manage quality by predicting the responses of individual workers.

While most crowdsourcing applications focus on relatively rote work such as basic image classification (*Schenk & Guittard, 2009*), many researchers have studied how to incorporate crowdsourcing into creative work. Some examples include the studies of *Bernstein et al. (2015)* and *Teevan, Iqbal & von Veh (2016)*, both of which leverage crowd workers for prose writing; *Kittur (2010)*, where the crowd helps with translation of poetry; *Chilton et al. (2013)*, where the crowd develops taxonomic hierarchies; and *Dontcheva, Gerber & Lewis (2011)*, where crowd workers were asked to ideate new and interesting applications or uses of common everyday objects, such as coins. In the specific context of ML, *Kaggle* provides a competition platform for *expert* crowd participants to create predictive models, allowing data holders to crowdsource predictive modeling, but prediction tasks are still designed by the data providers not the crowd.

In many crowdsourcing applications where workers contribute novel information, a propose-and-rank algorithm is typically used to ensure that crowd resources are focused on high-quality contributions by ranking those contributions in advance (*Siangliulue et al., 2015*; *Salganik & Levy, 2015*). For example, the "Wiki Surveys" project (*Salganik & Levy, 2015*) asks volunteers to contribute and vote on new ideas for improving quality-of-life in New York City. Wiki Surveys couples a proposal phase with a ranking and selection step, to create ideas and then filter and select the best ideas for the city government to consider. None of these studies applied crowdsourced creativity to problems of ML or data collection, however.

Two sets of studies are perhaps most closely related to the research here. The first focuses on non-experts who built ML tools that enable non-experts to build ML models, and investigated their goals, methods, and the challenges they encountered *Yang et al. (2018)*. *Yang et al. (2018)* performed an empirical study, conducting interviews and surveys of non-experts, and their study serves as an important complement to our work. To the best of our knowledge, our study here is the first experimental consideration of non-experts and their ability to design prediction tasks.

The second set of studies all consider the non-expert design of predictive features, that is, crowdsourced *feature engineering* (*Bongard et al., 2013*; *Bevelander et al., 2014*;

*Swain et al., 2015*; *Wagy et al., 2017*). This work studies the crowdsourcing of survey questions in multiple problem domains. Participants answered questions related to a quantity of interest to the crowdsourcer, such as how much they exercised vs. their obesity level or how much laundry they did at home compared with their home energy usage. Those participants were also offered the chance to propose new questions related to the quantity of interest (obesity level or home energy use). Algorithms were deployed while crowdsourcing occurred to relate answers to proposed questions (the features) to the quantity of interest, and thus participants were performing crowdsourced feature engineering with the goal of contributing novel predictive features of interest. Another similar study, Flock (*Cheng & Bernstein, 2015*), demonstrates that features built by non-experts working together with algorithms can improve supervised classifiers. However, these studies all limit themselves to feature engineering, and still require experts to design the supervised learning prediction task itself, that is, experts decide what is the quantity of interest to be predicted. Yet non-experts will be unable to apply ML to a new area of their own interest if they can only contribute features to pre-existing problems. Therefore, our work here generalizes this to ask individuals to design the *entire prediction task*, not just the features, by allowing non-experts to propose not only questions *related* to a quantity of interest, but also the quantity of interest *itself*.

## METHODS

Here, we describe our procedures for non-experts to design prediction tasks which are then ranked, categorized, and data are collected for top-ranked prediction tasks. After data are collected for a given task, supervised learning models are trained and assessed with cross-validation to estimate the learnability of the prediction task. The University of Vermont Institutional Review Board granted Ethical approval to carry out the study (determination number CHRBSS: 15-039). Collected data are available on Figshare (https://doi.org/10.6084/m9.figshare.9468512). Crowdsourcing experiment details, including sample sizes and rewards, are given in "Crowdsourcing Assignments".

### Prediction task ideation

To understand how non-experts can design and use ML, we introduce a protocol for the creation and data collection of supervised learning prediction tasks (several example prediction tasks are shown in Table 1). Inspired by "propose-and-rank" crowd ideation methods (Background), the protocol proceeds in three phases: (i) prediction task proposal, (ii) task selection by ranking, and (iii) data collection for selected tasks. Proposed prediction tasks may also be categorized or labeled by workers, allowing us to understand properties of proposed tasks. This is an end-to-end procedure in that crowd workers generate all prediction tasks and data without manual interventions from the crowdsourcer, allowing us to understand what types of topics non-expert workers tend to be interested in, and whether ML models can be trained on collected data to make accurate predictions.

#### Prediction task proposal

In the first phase, a small number of workers are directed to propose sets of questions (see Supplemental Materials for the exact wording of these and all instructions we used in our

**Table 1 Examples of non-expert-proposed prediction tasks.** Each task is a set of questions, one target and p inputs, all generated entirely by non-experts (Prediction Task Ideation). After crowdsourced ranking (Prediction Task Ranking and Selection) and data collection (Data Collection and Supervised Learning to Assess Learnability), answers to input questions form the data matrix **X** and answers to the target question form the target vector $y$. Machine learning algorithms try to predict the value of the target given only responses to the inputs. The learnability of the prediction task can then be assessed from the performance of these predictions. Prior work on crowdsourced feature engineering asks workers to contribute new predictive features (as input questions, in this case) for an expert-defined target. Here we ask workers to propose the entire prediction task not just the features.

| | Prediction task |
|---|---|
| Target: | **What is your annual income?** |
| Input: | You have a job? |
| Input: | How much do you make per hour? |
| Input: | How many hours do you work per week? |
| Input: | How many weeks per year do you work? |
| Target: | **Do you have a good doctor?** |
| Input: | How many times have you had a physical in the last year? |
| Input: | How many times have you gone to the doctor in the past year? |
| Input: | How much do you weigh? |
| Input: | Do you have high blood pressure? |
| Target: | **Has racial profiling in America gone too far?** |
| Input: | Do you feel authorities should use race when determining who to give scrutiny to? |
| Input: | How many times have you been racially profiled? |
| Input: | Should laws be created to limit the use of racial profiling? |
| Input: | How many close friends of a race other than yourself do you have? |

experiments). Workers are instructed to provide a prediction task consisting of one *target question* and $p = 4$ *input questions*. We focused on four input questions here to keep the proposal problem short; we discuss generalizing this in "Discussion". Several examples of tasks proposed by workers are shown in Table 1. Workers are told that our goal is to predict what a person's answer will be to the target question after only receiving answers to the input questions. Describing the prediction task proposal problem in this manner allows workers to envision the underlying goal of the supervised learning task without the need to discuss data matrices, response variables, predictors, or other field-specific vocabulary. Workers were also instructed to use their judgment and experience to determine "interesting and important" tasks. Importantly, *no examples of questions were shown to workers*, to help ensure they were not biased in favor of the example (we investigate this bias with a randomized trial; see "Randomized Trial for Assignment Design: Giving Examples" and "Randomized Trial: Giving Examples of Prediction Tasks"). Workers were asked to write their questions into provided text fields, ending each with a question mark. They were also asked to categorize the type of answer expected for each question; for simplicity, we directed workers to provide questions whose answers were either numeric or true/false (Boolean), though this can be readily generalized. Lastly, workers in the first phase are also asked to provide answers to their own questions.

### Prediction task ranking and selection

In the second phase, new workers are shown previously proposed tasks, along with instructions again describing the goal of predicting the target answer given the input answers, but these workers are asked to (i) rank the task according to our criteria (described below) but using their own judgment, and (ii) answer survey questions describing the tasks they were shown. It is useful to keep individual crowdsourcing tasks short, so it is generally too burdensome to ask each worker to rank all $N$ tasks. Instead, we suppose that workers will study either one task or a pair of tasks depending on the ranking procedure, complete the survey questions for the task(s), and, if shown a pair of tasks, to rate which of the two tasks they believed was "better" according to the instructions. To use these ratings to develop a global ranking of tasks from "best" to "worst", we apply top-$K$ ranking algorithms ("Ranking Proposed Prediction Tasks"). These algorithms select the $K$ most suitable tasks to pass along to phase three.

#### Task categorization

As an optional part of phase two, data can be gathered to categorize what types of tasks are being proposed, and what are the properties of those tasks. To categorize tasks, we asked workers what the topic of each task is, whether questions in the task were subjective or objective, how well answers to the input questions would help to predict the answer to the target question, and what kind of responses other people would give to some questions. We describe the properties of proposed tasks in "Results".

### Data collection and supervised learning to assess learnability

In phase three, workers were directed to answer the input and target questions for the tasks selected during the ranking phase. Workers could answer the questions in each selected task only once but could work on multiple prediction tasks. In our case, we collected data from workers until each task had responses from a fixed number of unique workers $n$, but one could specify other criteria for collecting data. The answers workers give in this phase create the datasets to be used for learning. Specifically, the $n \times p$ matrix $\mathbf{X}$ consists of the $n$ worker responses to the $p$ input questions (we can also represent the answers to each input question $i$ as a predictor vector $x_i$, with $\mathbf{X} = [x_1, \ldots, x_p]$). Likewise, the answers to the target question provide the supervising or target vector $y$.

   After data collection, supervised learning methods can be applied to find the best predictive model $\hat{f}$ that relates $y$ and $\mathbf{X}$, for example, $y = \hat{f}(\mathbf{X})$. Then learnability of the prediction task and data can be assessed by model performance using cross-validation, although this assessment depends on the predictive model used, and more advanced models may lead to improved learnability. In our case, we focused on random forests (*Breiman, 2001*) as our predictive model, a commonly used and general-purpose ensemble learning method. Random forests work well on both linear and nonlinear prediction tasks and can be used for both regressions (where $y$ is numeric) and classifications (where $y$ is categorical). However, any supervised learning method can be applied in this context. For hyperparameters used to fit the forests, we chose 200 trees per forest, a split criterion of MSE for regression and Gini impurity for classification, and tree nodes are expanded until

all leaves are pure or contain fewer than two samples. These are commonly accepted choices for hyperparameters, but of course careful tuning of these values (using appropriate cross-validation) can only result in better learning than we report here.

## Ranking proposed prediction tasks

Not all workers will propose meaningful tasks, so it is important to include a ranking step (phase two) that filters out low-quality (less meaningful) tasks while promoting high-quality (more meaningful) tasks.

To ground our ranking process, here we define a prediction task as "meaningful" if it is both important, as determined by crowd judgments and learnable, as assessed by cross-validation of a trained predictive model. A non-important prediction task may be one that leads to unimportant or unimpactful broader consequences, the target variable may not be worth predicting, or the task may simply recapitulate known relationships (*"Do you think 1 + 1 = 2?"*). As importance can be subjective, here we rely on the crowd to collectively certify whether a task is important or not according to their own criteria. Although it may be necessary to guide non-experts to specific areas of interest (see "Discussion"), here we avoid introducing specific judgments or criteria so that we can see what "important" prediction tasks are proposed by non-experts.

To be meaningful as a prediction task, the task must also be learnable. Indeed, another characteristic of poor prediction tasks is a lack of *learnability*, defined as the ability for a predictive model trained on data collected for the prediction task to accurately generalize to unseen data. For a binary classification task, for example, one symptom of poor learnability (but not the only issue) is an imbalance of the class labels. For example, the target question "Do you think running and having a good diet are healthy?" is likely to lead to very many "true" responses and very few "false" responses. Such data lacks diversity (in this case, in the labels), which makes learning difficult. Of course, while a predictive model in such a scenario is not especially useful, the relationships and content of the target and input questions are likely to be meaningful, as we saw in some of our examples; see Supplemental Materials. In other words, a predictive task can be about an important topic or contain important information, but if it is not learnable then it is not meaningful *as a prediction task*.

Here we detail how to use crowd feedback to efficiently rank tasks based on importance and learnability. We refer to the ratings given by the crowd as "perceived importance" and "perceived learnability" the reflect the subjective nature of their judgments. The outcome of this ranking (Result) also informs our investigation of RQ4. In the context of crowdsourcing prediction tasks, the choice of ranking criteria gives the crowdsourcer flexibility to guide workers in favor of, not necessarily specific types of tasks, but tasks that possess certain features or properties. This balances the needs of the crowdsourcer (and possible budget constraints) without restricting the free-form creative ideation of the crowd.

### Perceived importance ranking

We asked workers to use their judgment to estimate the "importance" of tasks (see Supplemental Materials for the exact wording of instructions). To keep workloads

manageable, we ask workers to compare two tasks at a time, with a simple "Which of these two tasks is more important?"-style question. This reduces the worker's assignment to a pairwise comparison. Yet, even reduced to pairwise comparisons, the global ranking problem is still challenging, as one needs $\mathcal{O}(N^2)$ pairwise comparisons for $N$ tasks, comparing every task to every other task. Furthermore, importance is generally subjective, so we need the responses of many workers and cannot rely on a single response to a given pair of tasks. Assuming we require $L$ independent worker comparisons per pair, the number of worker responses required for task ranking grows as $\mathcal{O}(LN^2)$.

Thankfully, ranking algorithms can reduce this complexity. Instead of comparing all pairs of tasks, these algorithms allow us to compare a subset of pairs to infer a latent score for each task, then rank all tasks according to these latent scores. For this work, we chose the following top-$K$ spectral ranking algorithm, due to *Negahban, Oh & Shah (2017)*, to rank non-expert-proposed tasks and extract the $K$ best tasks for subsequent crowdsourced data collection. The algorithm uses a comparison graph $G = (V, E)$, where the $N$ vertices denote the tasks to be compared, and comparison between two tasks $i$ and $j$ occurs only if $(i, j) \in E$. For our specific crowdsourcing experiment, we solicited $N = 50$ tasks during the proposal phase, so here we generated a single Erdös-Rényi comparison graph of 50 nodes with each potential edge exists independently with probability $p = 1.5\log(N)/N$ (this $p$ ensures $G$ is connected), and opted for $L = 15$. Increasing $L$ can improve ranking accuracy, but doing so comes at the cost of more worker time and crowdsourcer resources. The choice of an Erdős-Rényi comparison graph here is useful: when all possible edges are equally and independently probable, the number of samples needed to produce a consistent ranking is nearly optimal (*Negahban, Oh & Shah, 2017*).

### Perceived learnability ranking

As discussed above, prediction tasks lack learnability when (among other reasons) there is insufficient diversity in the dataset. If nearly every observation is identical, there is not enough "spread" of data for the supervised learning method to train upon; no trends will appear if every response to the input questions is identical or if every value of the target variable is equal. To avoid collecting data for tasks that are not learnable, and to assess how well non-experts can make determinations about learnability, we seek a means for workers to estimate for us the learnability of a proposed task when shown the input and target questions. The challenge is providing workers with an assignment that is, sufficiently simple for them to perform quickly yet does not require training or background in how supervised learning works.

To address this challenge, we designed an assignment to ask workers about their opinions of the set of answers we would receive to a given question (a form of meta-knowledge). We focused on a lack of diversity in the target variable. We also limited ourselves to Boolean (true/false) target questions, although it is straightforward to generalize to regressions (numeric target questions) by rephrasing the assignment slightly. Specifically, we asked workers what proportion of respondents would answer "true" to the given question. Workers gave a 1–5 Likert-scale response from (1) "No one will answer true" to (3) "About half will answer true" to (5) "Everyone will answer true". The idea is that, since a diversity of

responses is generally necessary (but not sufficient) for (binary) learnability, classifications that are balanced between two class labels are more likely to be learnable. To select tasks, we use a simple ranking procedure to seek questions with responses predominantly in the middle of the Likert scale. Specifically, if $t_{ij} \in \{1,\ldots,5\}$ is the response of the $i$-th worker to prediction task $j$, we take the aggregate learnability ranking to be

$$t_j = \left| 3 - \frac{\sum_{i=1}^{W} t_{ij}\delta_{ij}}{\sum_{i=1}^{W} \delta_{ij}} \right|, \tag{1}$$

where $W$ is the total number of workers participating in learnability ranking tasks, and $\delta_{ij} = 1$ if worker $i$ ranked task $j$, and zero otherwise. The closer a task's score is to 3, the more the workers agree that target answers would be evenly split between true and false, and so we rank tasks based on the absolute deviation from the middle score of 3. While Eq. (1) is specific to a 1–5 Likert scale variable, similar scores can be constructed for any ordinal variable.

This perceived learnability ranking task can be combined with a pairwise comparison methodology like the one described for perceived importance ranking. In our case, we elected to perform a simpler one-task assignment because perceived learnability ranking from Eq. (1) only requires examining the target question and because workers are less likely to need a relative baseline here as much as they may with perceived importance ranking, where a contrast effect between two tasks is useful for judging subjective values such as importance. Due to time and budget constraints we also took $K = 5$ for experiments using this ranking phase.

There are limitations of this approach to perceived learnability beyond focusing only on classifications. For one, soliciting this information does not address a potential lack of diversity in the **X** data, nor do we consider the perceived learnability of regressions. Likewise, it does not address myriad other issues that affect learnability, such as the capacity of the predictive model (*Vapnik, 2013*; *Valiant, 1984*). See "Discussion" for further discussion.

### Randomized trial for assignment design: giving examples

In addition to the crowdsourced proposal, ranking and data collection, we augmented our study with a randomized trial investigating the design of the prediction task proposal assignment (RQ3). Specifically, we investigated the role of providing an example of a prediction task.

Care must be taken when instructing workers to propose prediction tasks. Without experience in ML, they may be unable to follow instructions which are too vague or too reliant on ML terminology. Providing an example with the instructions is one way to make the assignment more clear while avoiding jargon. An example helps avoid the *ambiguity effect* (*Ellsberg, 1961*), where workers are more likely to avoid the task because they do not understand it. However, there are potential downsides as well: introducing an example may lead to *anchoring* (*Tversky & Kahneman, 1974*) where workers will be biased towards

**Table 2 Summary of crowdsourcing assignments.** Rewards in USD.

| Phase | Assignment | Reward | Responses | Workers |
|---|---|---|---|---|
| 1 | Prediction task proposal | $3.00 | 50 | 50 |
| 2 | Perceived importance rating and task categorization | $0.25 | 2,042 | 239 |
| 2 | Perceived learnability rating | $0.05 | 835 | 83 |
| 3 | Data collection, perceived importance tasks | $0.12 | 2,004 | 495 |
| 3 | Data collection, perceived learnability tasks | $0.12 | 990 | 281 |
| 1 | Prediction task proposal (randomized trial) | $3.00 | 90 | 90 |
| 2 | Task categorization (randomized trial) | $0.13 | 458 | 61 |

proposing tasks related to the example and may not think of important, but different prediction tasks.

Workers who did not participate in previous assignments were randomly assigned to one of three comparison groups or "arms" when accepting the prediction task proposal assignment (simple random assignment). One arm had no example given with the instructions and was identical to the assignment studied previously (Characteristics of Proposed Tasks). This arm serves as a baseline or control group. The second arm included with the instructions an example related to obesity (*An example target question is: "Are you obese?"*), and the third arm presented an example related to personal finance (*An example target question is: "What is your current life savings?"*). The presence or absence of an example is the only difference across arms; all other instructions were identical and, crucially, workers were not instructed to propose prediction tasks related to any specific topical domain or area of interest.

After we collected new prediction tasks proposed by workers who participated in this randomized trial, we then initiated a followup task categorization assignment (Prediction Task Ranking and Selection) identical to the categorization assignment discussed previously but with two exceptions: we asked workers to only look at one prediction task per assignment and we did not use a comparison graph as here we will not rank these tasks for subsequent data collection. The results of this categorization assignment allow us to investigate the categories and features of the proposed prediction tasks and to see whether or not the tasks differ across the three experimental arms.

# RESULTS

## Crowdsourcing assignments

We performed our experiments using Amazon Mechanical Turk during August 2017. Assignments were performed in sequence, first prediction task proposal (phase one), then ranking and categorization (phase two), then data collection (phase three). These assignments and the numbers of responses and numbers of workers involved in each are detailed in Table 2, as are the rewards workers were given. Rewards were determined based on estimates of the difficulty or time spent on the assignment, so proposing a prediction task had a much higher reward ($3 USD) than providing data by answering the

task's questions ($0.12 USD). No responses were filtered out at any point, although a small number of responses (less than 1%) were not successfully recorded.

We solicited $N = 50$ prediction tasks in phase one, compensating Mechanical Turk workers $3 for the assignment. Workers could submit only one task. A screenshot of the assignment interface for this phase (and all phases) is shown in the Supplemental Materials. Some example prediction tasks provided by crowd workers are shown in Table 1; all 50 tasks are shown in the Supplemental Materials. After these tasks were collected, phase two began where workers were asked to rate the tasks by their perceived importance and perceived learnability and to categorize the properties of the proposed tasks. Workers were compensated $0.25 per assignment in phase two and were limited to examining at most 25 tasks total. After the second phase completed, we chose the top-10 most perceived important tasks and the top-five most perceived learnable tasks (Ranking Proposed Prediction Tasks) to pass on to data collection (phase three). We collected data for these tasks until $n = 200$ responses were gathered for each prediction task (we have slightly less responses for some tasks as a few responses were not recorded successfully; no worker responses were rejected). Workers in this phase could respond to more than one task but only once to each task.

For the randomized trial on the effects of providing an example prediction task (Randomized Trial for Assignment Design: Giving Examples), we collected $N = 90$ proposed prediction tasks across all three arms (27 in the no-example baseline arm, 33 in the obesity example arm, and 30 in the savings example arm), paying workers as before. We then collected 458 task categorization ratings, gathering ratings from five or more distinct workers per proposed task (no worker could rate more than 25 different prediction tasks). Since only one task was categorized per assignment instead of two, workers were paid $0.13 per assignment instead of the original $0.25 per assignment.

## Characteristics of proposed tasks

We examined the properties of prediction tasks proposed by workers in phase one (RQ2). We measured the prevalence of Boolean and numeric questions. In general, workers were significantly in favor of proposing Boolean questions over numeric questions. Of the $N = 50$ proposed tasks, 34 were classifications (Boolean target question) and 16 were regressions (numeric target question). Further, of the 250 total questions provided across the $N = 50$ tasks, 177 (70.8%) were Boolean and 73 were numeric (95% CI on the proportion of Boolean: 64.74–76.36%), indicating that workers were significantly in favor of Boolean questions over numeric. Likewise, we also found an association between whether the input questions were numeric or Boolean given the target question was numeric or Boolean. Specifically, we found that prediction tasks with a Boolean target question had on average 3.12 Boolean input questions out of 4 (median of four Boolean input questions), whereas tasks with a numeric target question had 2.31 Boolean input questions on average (median of two Boolean input questions). The difference was significant (Mann–Whitney test: $U = 368.5$, $n_{bool} = 34$, $n_{num} = 16$, $p < 0.02$). Although it is difficult to draw a strong conclusion from this test given the amount of data we have (only $N = 50$ proposed prediction tasks), the evidence we have indicates that workers tend

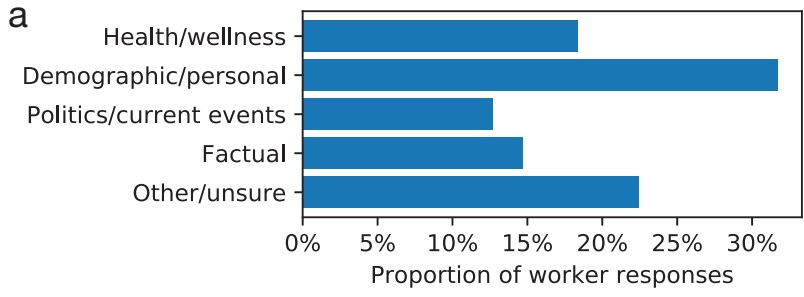

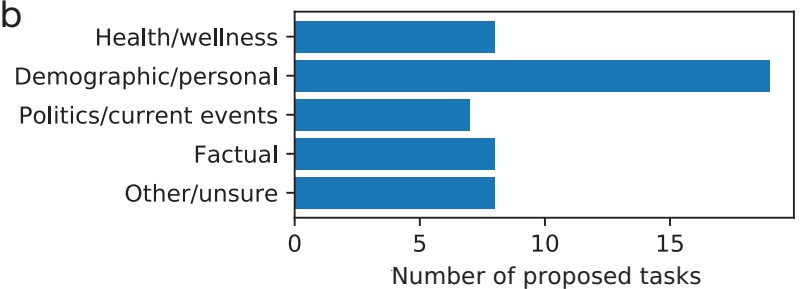

**Figure 1 Topical categories of proposed prediction tasks.** (A) Proportion of worker responses. (B) Number of proposed tasks. Panel B counts the majority categorization of each task.

to think of the same type of question for both the target and the inputs, despite the potential power of mixing the two types of questions.

To understand more properties of the questions workers proposed, we asked workers to categorize prediction tasks by giving survey questions about the tasks as part of the perceived importance rating assignment. We used survey questions about the topical nature or domain of the task (Fig. 1), whether the inputs were useful at predicting the target (Fig. 2), and whether the questions were objective or subjective (Fig. 3). Prediction task categories (Fig. 1) were selected from a multiple choice categorization we determined manually. Tasks about demographic or personal attributes were common, as were political and current events. Workers generally reported that the inputs were useful at predicting the target, either rating "agree" or "strongly agree" to that statement (Fig. 2). Many types of tasks were mixes between objective and subjective questions, while tasks categorized as "factual" tended to contain the most objective questions and tasks categorized as "other/unsure" contained the most subjective questions, indicating a degree of meaningful consistency across the categorization survey questions.

When ranking classification tasks by perceived learnability, we asked workers about the diversity of responses they expected others to give to the Boolean target question, whether they believed most people would answer false to the target question, or answer true, or if people would be evenly split between true and false (Fig. 4). We found that generally there was a bias in favor of positive (true) responses to the target questions, but that workers felt that many questions would have responses to the target questions be split between true and false. This bias is potentially important for a crowdsourcer to consider when designing her own tasks, but seeing that most Boolean target questions are well split
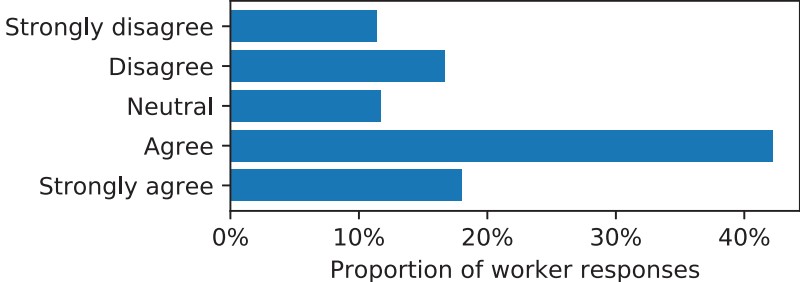

**Figure 2** Worker responses to, "Are the input questions useful at predicting answers to the target question?" when asked to categorize proposed prediction tasks.

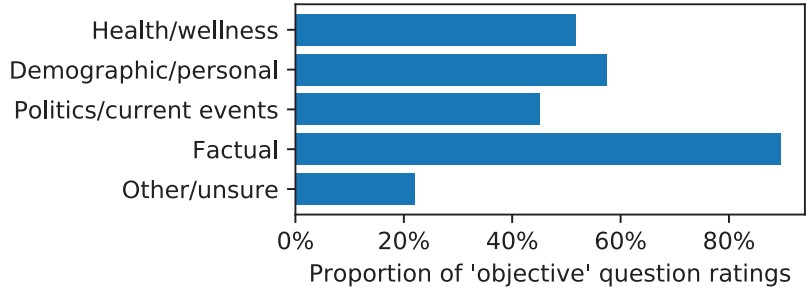

**Figure 3** Proportion of question ratings of "objective" instead of "subjective" vs. the majority category of the prediction task.

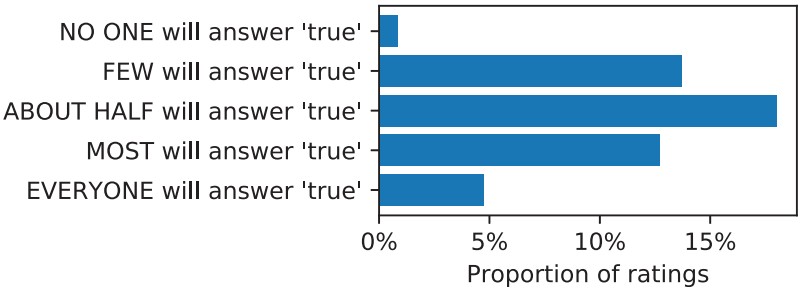

**Figure 4** Categorized diversity of the (Boolean) target questions.

between true and false response also supports that workers are proposing useful tasks; if the answer to the target question is always false, for example, then the input questions are likely not necessary, and the workers generally realize this when proposing their tasks.

## Prediction task learnability: supervised learning on collected data

Given the proposed prediction tasks and the selection of tasks for subsequent data collection, it is also important to quantify predictive model performance on these tasks (RQ5), allowing us to determine if the tasks are learnable given the data and model used. Since workers are typically not familiar with supervised learning, there is a risk they may be unable to propose learnable prediction tasks. At the same time, however, workers may not be locked into traditional modes of thinking, such as assumptions that predictors

are linearly related to the target, leading to interesting and potentially unexpected combinations of predictor variables and the response variable.

Here we trained and evaluated random forest regressors and classifiers (Data Collection and Supervised Learning to Assess Learnability), depending on whether the proposer flagged the target question as either numeric or Boolean, using the data collected for the 15 selected prediction tasks. Learnability (predictive performance) was measured using the coefficient of determination for regressions and mean accuracy for classifications, as assessed with $k$-fold cross-validation (stratified $k$-fold cross-validation if the task is a classification). To assess the variability of performance over different datasets, we used bootstrap replicates of the original crowd-collected data to estimate a distribution of cross-validation scores. There is also a risk that class imbalance may artificially inflate performance: when nearly every target variable is equal always predicting the majority class label can appear to perform well. To account for class imbalance, we also trained on a shuffled version of each task's dataset, where we randomly permuted the rows of the data matrix $\mathbf{X}$, breaking the connection with the target variable $y$. If models trained on these data performed similarly to models trained on the real data, then it is difficult to conclude that learning has occurred, although this does not mean the questions are not meaningful, only that the data collected does not lead to useful predictive models.

The results of this learnability assessment procedure are shown in Fig. 5. We quantify the practical effect size with Cohen's $d$ comparing the real training data to the shuffled control, and in Fig. 5 we highlight in green any tasks with Cohen's $d > 2$. Many of the 10 perceived importance-ranked tasks in Figs. 5A–5J demonstrate a lack of learnability but at least two of the ten perceived important tasks, one regression and one classification, also show significant learning[1]. At the same time, four out of the five perceived learnability-ranked tasks (Figs. 5K–5O) showed strong predictive performance, further indicating the ability of non-experts to perform learnability assessments.

These results show that, while many of the worker-proposed prediction tasks are difficult to learn on, and caution must be taken to instruct non-experts about the issue of class imbalance, it is possible to generate tasks where learning can be successful and to assess this with an automatic procedure such as testing the differences of the distributions shown in Fig. 5. Further, for the classification tasks in Figs. 5K–5O, by assessing learnability (predictive performance) using cross-validation, we can compare with non-expert judgments of perceived learnability. Despite perceived learnability being oversimplified (only considering the issue of class imbalance), we found that most perceived learnable tasks demonstrated learnability in practice.

## Randomized trial: giving examples of prediction tasks

To understand what role an example may play—positive or negative—in task proposal, we conducted a randomized trial investigating the instructions given to the workers. As described in "Randomized Trial for Assignment Design: Giving Examples", we conducted a three-armed randomized trial. Workers who did not participate in the previous study were asked to propose a prediction task with instructions that either contained no example (baseline arm), contained an example related to obesity (obesity

[1] One regression showed poor performance scores for reasons we detail in the discussion.

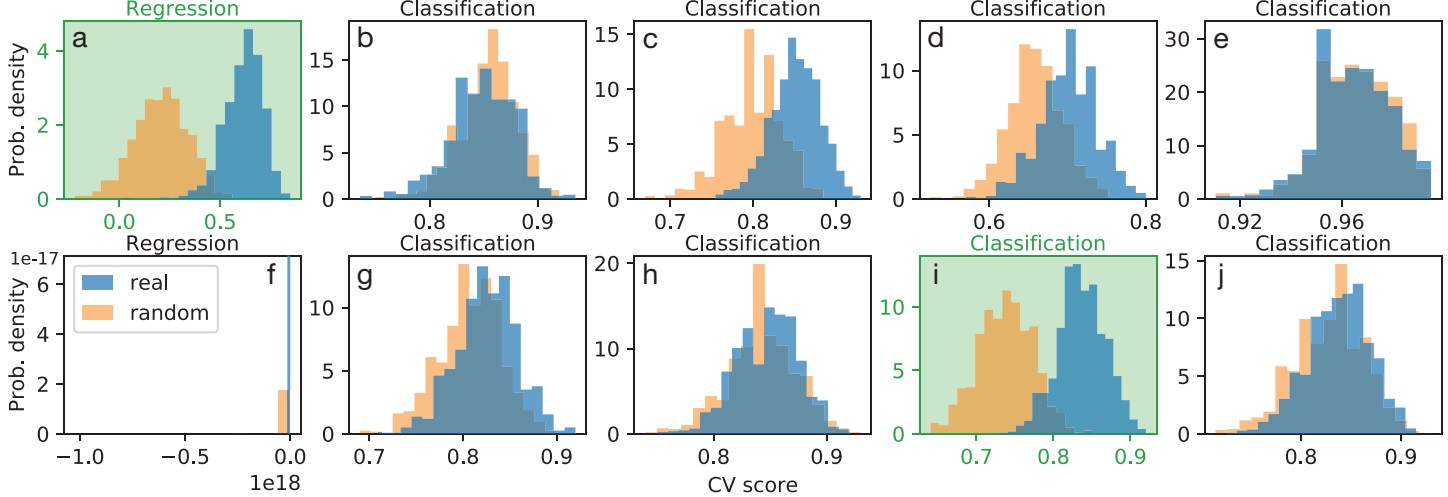

(a–j) Top perceived importance problems

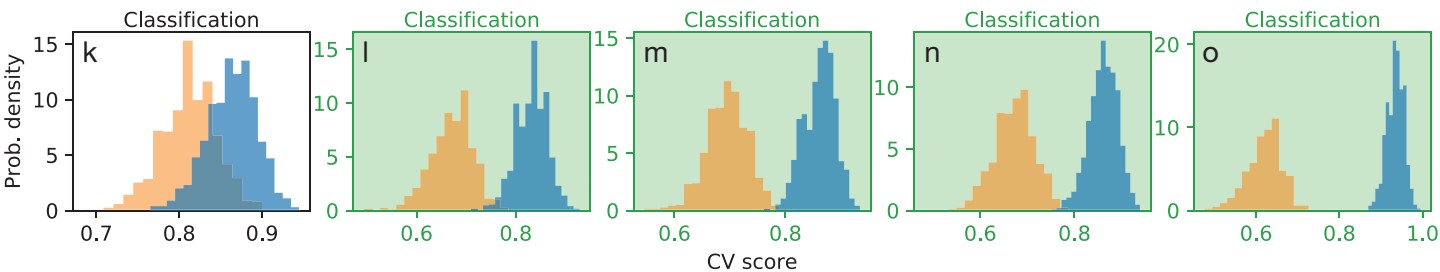

(k–o) Top perceived learnability (classification) problems

**Figure 5 Cross-validation scores for (A–J) the top-10 perceived importance ranked prediction tasks (Perceived Importance Ranking) and (K–O) the top-five perceived learnable prediction tasks (Perceived Learnability Ranking).** Performance variability was assessed with bootstrap replicates of the crowdsourced datasets and class imbalance was assessed by randomizing the target variable relative to the input data. At least two of the perceived importance tasks and four of the perceived learnable tasks—highlighted in green—demonstrate significant learnability (predictive performance) over random (Cohen's $d > 2$). Note that the regression task in (F) showed poor predictive performance for reasons we describe in the discussion.

arm), or contained an example related to personal savings (savings arm). The prediction tasks proposed by members of these arms were categorized and rated and from these ratings we study changes in task category, usefulness of input questions at answering the target question, if the questions are answerable or unanswerable, and if the questions are objective or subjective, as judged by workers participating in the followup rating tasks.

The results of this trial are summarized in Fig. 6 and Tables 3 and 4. We found that:

- Prediction task categories changed due to the examples (Fig. 6), with more "demographic/personal" tasks, fewer "politics/current events", and fewer "factual" questions under the example treatments compared with the baseline. This change was significant ($\chi^2 = 52.73$, $p < 0.001$).

- Workers shown the savings example were significantly more likely than workers in other arms to propose questions with numeric responses instead of Boolean responses: 60% of questions proposed in the savings arm were numeric compared with 25% in the no-example baseline (Fisher exact, $p < 0.001$).

- All three arms were rated as having mostly answerable questions, with a higher proportion of answerable questions for both example treatments: 92% of ratings were 'answerable' for both example treatments compared with 82% for the baseline (Table 3). Proportions for both example treatments were significantly different from the baseline (Fisher exact, $p < 0.02$).

- Workers more strongly agreed that the inputs were useful at predicting the target for prediction tasks proposed by workers under the example treatments than the tasks proposed under the no-example baseline. The overall increase was not large, however, and tested as significant ($\chi^2 = 16.35$, $p < 0.005$) only for the savings example vs. the baseline.

- Questions proposed under the example treatments were more likely to be rated as objective than questions proposed under the no-example baseline: 67% and 68% of ratings were "objective" for the obesity and savings examples, respectively, compared with 59% for the baseline (Table 3). However, this difference was not significant for either treatment (Fisher exact, $p > 0.05$).

Taken together, the results of this experiment demonstrate that examples, while helping to explain the assignment, will lead to significant changes in the features and content of proposed prediction tasks. Individuals may provide better and somewhat more specific questions, but care may be needed when selecting which examples to use, as individuals may potentially anchor onto those examples in some ways when designing their own prediction tasks. Such anchoring may be undesired but it may also be useful at "nudging" non-experts towards tasks related to a problem area of interest; see discussion for more.

## DISCUSSION

Here we studied the abilities of non-experts to independently design supervised learning prediction tasks. Recruiting crowd workers as non-experts, we determined that non-experts were able to propose important or learnable prediction tasks with minimal instruction, but several challenges demonstrate that care should be taken when developing instructions as non-experts may propose trivial or "bad" tasks. Analyzing the proposed prediction tasks, we found that non-experts tended to favor Boolean questions over numeric questions, that input questions tended to be positively correlated with target questions, and that many prediction tasks were related to demographic or personal attributes.

Learnability is estimated using cross-validation (Fig. 5). This estimate encompasses the prediction task, the predictive model, and the training data. However, we also ask non-experts to estimate learnability. Can non-experts do this without knowing about cross-validation? Our answer is yes, for the case of classification. Despite class imbalance, our measure of perceived learnability, being only one factor affecting learnability, 4 out of 5

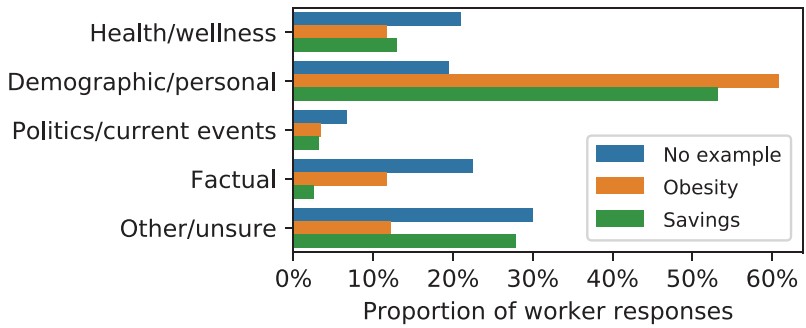

**Figure 6 Categories of proposed prediction tasks under the different instructional treatments.** Tasks proposed by workers who saw either example were more likely to be rated as demographic or personal and less likely to be considered factual. Interestingly, the obesity example led to fewer proposed tasks related to health or wellness.

**Table 3 Typical ratings and features of prediction tasks proposed under the three instruction types (the no-example baseline, the obesity example, and the savings example).**

| Rating or feature (variable type) | Mean $x$ baseline | Mean $x$ obesity | Mean $x$ savings |
|---|---|---|---|
| Perceived task importance ($x$ = 1–5 Likert; 5: Strongly agree) | 3.09 | 3.16 | **3.50** |
| Inputs are useful ($x$ = 1–5 Likert; 5: Strongly agree) | 3.36 | **3.91** | **3.67** |
| Questions are answerable ($x$ = 1) or unanswerable ($x$ = 0) | 0.82 | **0.92** | **0.92** |
| Questions are objective ($x$ = 1) or subjective ($x$ = 0) | 0.59 | 0.67 | 0.68 |
| Questions are numeric ($x$ = 1) or Boolean ($x$ = 0) | 0.25 | 0.35 | **0.60** |

Note:
 Bold treatment quantities show a significant difference ($p < 0.05$) from the baseline (Chi-square tests for Likert $x$; Fisher exact tests for binary $x$).

**Table 4 Statistical tests comparing the categories and ratings given for prediction tasks generated under the no-example baseline with the categories and ratings of tasks generated under the obesity example and savings example baselines.**

| Difference in | Test | Baseline vs. obesity | | Baseline vs. savings | |
|---|---|---|---|---|---|
| | | Statistic | $p$-Value | Statistic | $p$-Value |
| Task categories (see Fig. 6) | Chi-square | 52.73* | $<10^{-10}$ | 52.73* | $<10^{-9}$ |
| Perceived task importance (see Table 3) | Chi-square | 8.57 | $>0.05$ | 11.84* | $<0.02$ |
| Inputs are useful (see Table 3) | Chi-square | 16.35* | $<0.005$ | 7.20 | $>0.1$ |
| Answerable/unanswerable (see Table 3) | Fisher exact | —* | 0.0083 | —* | 0.012 |
| Objective/subjective (see Table 3) | Fisher exact | — | 0.12 | — | 0.078 |
| Numeric/Boolean (see Table 3) | Fisher exact | —* | 0.041 | —* | $<10^{-8}$ |

Notes:
 * $p < 0.05$.
 For categorical and Likert-scale ratings we used a Chi-square test of independence while for binary ratings we used a Fisher exact test.

of the top-rated "learnable" tasks demonstrated learnability using cross-validation. Meanwhile, two of the ten top-rated "important" tasks also demonstrated learnability, indicating that non-experts can generate meaningful prediction tasks.

It is worth speculating on the origins of the observed prediction task properties. For instance, the crowd workers could favor Boolean questions because they can be proposed more quickly (due to being less cognitively demanding) and crowd workers wish to work quickly to maximize their earnings. Or they could favor Boolean questions because their prior experience leaves them less familiar with numerical quantities. Likewise, a focus on demographic or personal attributes within prediction tasks could reflect the inherent interests of the participants or could be due to influences from prior work on the crowdsourcing platform. More generally, a useful future direction of study is to better understand the backgrounds of the non-experts. For example, how is the prediction task associated with the prior experience, domain expertise or education level of the non-expert who proposed the task?

To better understand how framing the problem of designing a prediction task may affect the tasks workers proposed, we also conducted a randomized trial comparing tasks proposed by workers shown no example to those shown examples, and found that examples significantly altered the categories of proposed prediction tasks. These findings demonstrate the importance of carefully considering how to frame the assignment, but they also reveal opportunities. For example, it is less common for non-expert workers to mix Boolean and numeric questions, but workers that do propose such mixtures may be identified early on and then steered towards particular tasks, perhaps more difficult tasks. Likewise, given that examples have a powerful indirect effect on prediction task design, examples may be able to "nudge" non-experts in one direction while retaining more creativity than if the non-experts were explicitly restricted to designing a particular type of prediction task. We saw an example of this in "Randomized Trial: Giving Examples of Prediction Tasks": non-expert workers shown the savings example were over 2.5 times more likely to propose numeric questions than workers shown no example.

Our experiments have limitations that should be addressed in future work. For one, we only considered non-experts on the Mechanical Turk platform, and work remains to see how our results generalize beyond that group (*Berinsky, Huber & Lenz, 2012*; *Chandler et al., 2019*). It is also important to explore more ML methods than we used here to learn predictive models on non-expert prediction tasks, especially as ML is a rapidly-changing field. Indeed, learnability can only be improved upon with more advanced methods. Further, assessing learnability only through judgments of class imbalance is an oversimplification of the issues that can affect learnability, although this oversimplification helped enable non-experts to judge learnability, at least in a limited manner, as shown in Fig. 5. Likewise, our ranking procedure considered the perceived importance and perceived learnability of prediction tasks separately, yet the most meaningful tasks should be ranked highly along both dimensions. With our prediction task proposal framework, we limited non-experts to numeric or Boolean questions, and a total of five questions per prediction task, but varying the numbers of questions and incorporating other answer types are worth exploring. For numeric questions, one important consideration is the choice of *units*. Indeed, we encountered one regression task (mentioned previously; see Supplemental Materials) where learning failed because the questions involved distances and volumes, but workers were not given information on units, leading to wildly varying answers. This teaches us that

non-experts may need to be asked if units should be associated with the answers to numeric questions when they are asked to design a prediction task.

Our experiment procedures were used to address this study's research questions, but in the context of crowdsourcing, our end-to-end propose-and-rank procedure for soliciting prediction tasks can serve as an efficient crowdsourcing algorithm. To avoid wasting resources on low-quality or otherwise inappropriate prediction tasks, efficient task selection and data collection algorithms are needed to maximize the ability of a crowd of non-experts to collectively generate suitable prediction tasks while minimizing the resources required from a crowdsourcer. When allowing creative contributions from a crowd, a challenge is that workers may propose trivial or uninteresting tasks. This may happen intentionally, due to bad actors, or unintentionally, due to workers misunderstanding the goal of their assignment. Indeed, we encountered a number of such proposed prediction tasks in our experiments, further underscoring the need for both careful instructions and the task ranking phase. Yet, we found that the task ranking phase did a reasonable job at detecting and down-ranking such prediction tasks, although there is room to improve on this further, for example, by reputation modeling of workers or implementing other quality control measures (*Allahbakhsh et al., 2013*; *Scholer, Turpin & Sanderson, 2011*; *Lease, 2011*) or perhaps by providing dynamic feedback earlier, when tasks are being proposed. More generally, it may be worth combining the ranking and data collection phases, collecting data immediately for all or most prediction tasks but simultaneously monitoring the tasks as data are collected for certain specifications and then dynamically allocating more incoming workers to the most suitable subset of prediction tasks (*Li et al., 2016*; *McAndrew, Guseva & Bagrow, 2017*).

Allowing the crowd to propose prediction tasks requires more work from the researcher on prediction task specification than in traditional applied ML. Traditionally, considerable effort is placed on model fitting, model validation, and feature engineering. AutoML will become increasingly helpful at model fitting and model validation while non-experts contributing predictive features may take some or even all of the feature engineering work off the researcher's hands. If crowd-proposed tasks are used, the researcher will need to consider how best to specify prediction task requirements. While here we allowed the crowd to ideate freely about tasks, with the goal of understanding what tasks they were most likely to propose, in practice a researcher is likely to instead focus on particular types of prediction tasks. For example, a team of medical researchers or a team working at an insurance firm may request only prediction tasks focused on health care. Future work will investigate methods for steering the crowd towards topics of interest such as health care, in particular on ways of focusing the crowd while biasing workers as little as possible.

Many interesting, general questions remain. For one, more investigation is needed into how non-experts work on ML prediction tasks. Which component or step of designing a prediction task is most challenging for non-experts? What aspects of ML, if any, are most important to teach to non-experts? Can studies of non-experts help inform teaching methodologies for turning ML novices into experts? Likewise, can teaching methodologies for learning about ML inform better ways to help non-experts contribute to ML?

## CONCLUSION

In this study, we investigated how well non-experts, individuals without a background in ML, could contribute to ML. While non-experts have long contributed training data to power pre-defined ML tasks, it remains unclear whether and to what extent non-experts can apply existing ML methods in new problem areas. We asked non-experts to design their own supervised learning prediction tasks, then asked other non-experts to rank those tasks according to criteria of interest. Finally, training data were collected for top-ranked tasks and ML models were fit to those data. We were able to demonstrate that performant models can be trained automatically on non-expert tasks. We also studied the characteristics of proposed tasks, finding that many tasks were focused on health, wellness, demographics, or personal topics, that numeric questions were less common than Boolean questions, and that there was a mix of both subjective and objective questions, as rated by crowd workers. Using a randomized trial on the effects of instructional messages showed that simple examples caused non-experts to change their approaches to prediction tasks: for example, non-experts shown an example of a prediction task related to personal finance were significantly more likely to propose numeric questions.

In general, the more that non-experts can contribute creatively to ML, and not merely provide training data, the more we can leverage areas such as AutoML to design new and meaningful applications of ML. More diverse groups can benefit from such applications, allowing for broader participation in jobs and industries that are changing due to machine-learning-driven workplace automation.

## ACKNOWLEDGEMENTS

Any opinions, findings, and conclusions or recommendations expressed in this material are those of the author(s) and do not necessarily reflect the views of the funders.

### Funding

This material is based upon work supported by the National Science Foundation under Grant No. IIS-1447634 and by Google Open Source under the Open-Source Complex Ecosystems And Networks (OCEAN) project. The funders had no role in study design, data collection and analysis, decision to publish, or preparation of the manuscript.

### Grant Disclosures

The following grant information was disclosed by the authors:
National Science Foundation: IIS-1447634.
Open-Source Complex Ecosystems And Networks (OCEAN).

### Competing Interests

The authors declare that they have no competing interests.

## Author Contributions

- James P. Bagrow conceived and designed the experiments, performed the experiments, analyzed the data, performed the computation work, prepared figures and/or tables, authored or reviewed drafts of the paper, and approved the final draft.

## Ethics

The following information was supplied relating to ethical approvals (i.e., approving body and any reference numbers):

The University of Vermont granted Ethical approval to carry out the study (Determination Number CHRBSS: 15-039).

## Data Availability

Data are available on Figshare: Bagrow, James (2020): Democratizing AI: Non-expert design of prediction tasks. figshare. Dataset. DOI 10.6084/m9.figshare.9468512.v1.

## Supplemental Information

Supplemental information for this article can be found online at http://dx.doi.org/10.7717/peerj-cs.296#supplemental-information.

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
