# Peer review of "Democratizing AI: non-expert design of prediction tasks"

_PeerJ Computer Science, doi:10.7717/peerj-cs.296_

## Round 0.1 · original submission · Major Revisions

I have received comments from two reviewers. Both of them are very positive about the research and contributions presented in this article, as you can see. There are however a number of major revisions that have been recommended to be carried out before the paper can be considered for acceptance. There are suggestions for major revisions particularly from reviewer 1, who suggests to be clearer with the structure and presentation of the research questions, adding further definitions where appropriate, as well as further discussing, justifying and validating some of the findings.

Reviewer 1 ·

Basic reporting

Overall, the article is written in professional language with minimal typos. Supplementary materials are well-appreciated as they were critical for understanding the experimental design. There are several areas that require additional clarification, restructuring, and/or modifications.

There are several terminologies and concepts in the article such as “meaningful task”, “interesting task”, “learnability”, “ML application” (as opposed to ML model), “creative task”, “interesting task” and “task importance” which are sprinkled throughout the article with clear definition and subsequently lack clear operationalization of those concepts. Overall, I would encourage the author to pay attention to defining all concepts (especially those mentioned in research questions and those that were used in analysis) and provide sufficient background in their relevance before using them in methods and analysis. For example, the research question 1 asks if non-ML experts can design “meaningful” prediction tasks. What does it mean for a task to be meaningful (and for whom or from whose perspective) and how do you measure meaningfulness of a task? The author hints at importance and learnability as two ways to potentially measure meaningfulness of tasks, and it would be great if the authors made well-articulated arguments as to why those two concepts would be associated with meaningfulness. Learnability may be a well-known concept in machine learning community, but its definition is debatable. Since task learnability is a significant portion of the analysis, I would encourage authors to provide some background into its definition and why it is relevant for measuring meaningfulness. “Prediction tasks lack learnability when there is insufficient diversity in the dataset” is also very confusing. I think the author means class balance rather than actual diversity.

I would also encourage the author to make direct connections between all research questions proposed and their corresponding methods, analysis, and results. Without these anchors, the results are difficult to follow. This could be achieved by labeling research questions and pointing to the appropriate research questions in the subsequent sections or organizing the sections to directly align with the research questions. There’s also an abrupt change in topic between sections 3-4 and section 5. It seems like the article consists of two studies, but one of them has dedicated methods and results sections while the other has results as a subsection. I would encourage reorganizing the structure to clearly delineate separate sections and to have a common pattern of presenting the study such that the readers can easily link different results to different research questions and study designs.

Abstract could also be restructured into: (1) Problem or need statement, (2) Current state of the art, (3) Summarization of research goal, (4) Research design/methods/analyses, (5) Summary of results, (6) Summary of implications if available. Currently, the last sentence in the abstract paints a broad picture of the problem or need and is not directly connected to how this study has addressed the issue at hand.

There are a few suggestions for related works:

1. This article tries to examine non-experts’ ability to design prediction tasks. What’s critical to grounding this study is in discussing why it’s important for non-experts to design prediction tasks. The related works in this article is a bit lacking on this aspect while it has a lot of materials on crowd-based tasks related to ML. Why is it insufficient for the crowd/people to provide features? In motivating why non-ML experts (i.e., those with domain knowledge but no ML knowledge) need to be able to design ML tasks, I would encourage referring to publications on machine teaching by https://www.microsoft.com/en-us/research/group/machine-teaching-group/#!publications

2. Another missing background relevant to this article is a discussion around the process of designing a task by ML experts or non-ML experts and the challenges surrounding them. A well-designed process for non-experts should also work for ML experts. Does the MTurk task that the author has designed work equally well for ML experts? Which component or step of designing an ML task is most challenging for non-experts? The challenges that non-experts face must also be the same challenges that novice ML practitioners have faced before becoming ML experts. The article tries to answer this question by looking at experimenting with providing examples (i.e., priming), restricting inputs to booleans and numbers (i.e., representation), providing balanced set of labels, etc. I would encourage the authors to look at prior literature on the practice of ML (e.g., https://homes.cs.washington.edu/~pedrod/papers/cacm12.pdf) to draw connections to the current research questions and experimental designs.

3. The author talks about the similarity of the current study to existing ranking algorithms and tasks. I would encourage first discussing why ranking is relevant here and then discussing similar examples.

Experimental design

The authors propose importance and learnability ranking to determine if a prediction task is meaningful. Although the mathematical formulation and the subjective rating of determining importance and learnability seems valid, it is unclear why such a formulation is useful. This goes back to my prior comment about clarity of concept definitions, but perhaps more importantly the readers need to understand why learnability leads to meaningfulness and why that matters. Because learnability in this article is rather mathematical (can you separate the two classes) than practical (can you design a meaningful task), its relevance to non-experts is a bit lost in the analysis. Is a learnable task necessarily a meaningful task and an easy task for non-experts? It is difficult to evaluate experimental design because the narratives that support the design are weak, but perhaps there is a way to pivot these. Currently, there are bits and pieces but they are not threaded to tell a cohesive and compelling story.

The design of asking crowdworkers to propose a prediction task was clever in that the structure of target vs input questions and the types of data that they are working with helps workers think about prediction tasks more systematically by providing a framework. In my opinion, this is an important contribution for helping non-experts design prediction tasks that needs further exploration (i.e., without the structure, can non-experts design prediction tasks at all?). From that, learnability, importance, task characteristics analysis becomes the evaluation and description of the tasks generated with or without the structure. If the goal of the article was to help non-experts design tasks, then I would encourage the authors to consider organizing experiments into 3 contributions: (1) what tools (i.e., structure + examples) help non-experts (this also needs a baseline), (2) how to measure/characterize the quality of prediction tasks, (3) potential priming effect of examples. Methods and results were thoroughly explained. So threading the experiments in a clear narrative is encouraged.

Validity of the findings

The author did a great job detailing the findings and making sure the results are based on solid analysis and reproducible. I would have liked to see deeper dive in the discussions. For example, non-experts favored Boolean questions, but there needs to be some discussion on why. With ranking algorithm, there is an opportunity to provide feedback that helps with making prediction tasks “better” interactively. The second to last paragraph in discussion (442-448) needs some rewriting because it is unclear why researchers need to take their workload off of feature engineering or model fitting or what is exactly the issue of medical researchers requesting only prediction tasks focused on health care.

Additional comments

I believe the experiments and results are valid, but without a clear articulation of why, the article fails to deliver a convincing narrative and actionable takeaways. I would encourage taking a step back and asking the "why" question in putting the pieces together. I would also make sure to clearly state your research questions with all variables and concepts clarified and defined and making sure methods and results well align with the research questions.

·

Basic reporting

The paper is very well written and uses a clear and comprehensible style of writing. The ductus is scientific and technically correct.
The cited literature is sufficient and provides a good overview of the the topic.
Figures are suited to convey the intended message and are easy to read. Tables follow the standards in the field and raw data is shared in fighsare.
The article is understandable on its own, although some figures from the supplemental material could also be part of the original article (the preview of the survey form).
The document has some minor orthographic errors that need fixing.

E.g.
l. 206: Yet, (missing comma)
l. 118: Here, (missing comma)

The author uses the term "three-armed" experiment to indicate three different branches in a randomized controlled trial. This was unusual to see, but seems reasonable when reading the whole document. Maybe a figure showing the procedure in a flow chart would help understanding this more easily.

Experimental design

The set of experiments address a well-defined research question using crowdsourced participants from amazon turk. The experiments are well designed to address the research question, which is both relevant and meaningful.
The experiment addresses a research gap derived from literature and proposes a procedure to identify new ML-tasks from non-experts.
The methods used are suitable to answer the analytical questions and conducted with ethical review. Since data and methods are available and clearly defined, replication should be realizable by other investigators.

Validity of the findings

The findings are well documented and reflect both results and literature. Experimental effects are controlled for by a three-way randomized controlled trial. Conclusions address the problem statements from the introduction. Limitations are clearly defined and address a challenge with the experiment. Improvements for future research are clearly outlined.

I found only one aspect lacking from the whole research design was domain expertise and user characteristics. It may be interesting to see level of education and domain expertise from the sample. However, the findings are relevant without this information as well.

Additional comments

I was positively surprised with this manuscript, as it addresses a research area that is insufficiently studied. The proposed process seems reasonable, however it may need additional tailoring for more domain-specific and relevant tasks. Good work!

---

## Round 0.2 · Minor Revisions

Reviewers appreciate the improved revision as well as the detailed response to the reviews. While they are both supportive of this paper being ultimately accepted, they have both recommended a set of minor revisions that they paper should address prior to publication. These minor revisions mainly refer to clarity and detail in some of the parts of the paper. Reviewers have clearly listed the suggested revisions, which you can find in the attached reviews.

Reviewer 1 ·

Basic reporting

First of all, I thank the authors for such thoughtful and detailed responses to the questions I raised. I am very pleased with the responses and the revision. I have minor suggestions for clarity and strengthening of the arguments, which might be subjective so bare with me.

The authors’ responses were direct, thoughtful, and clear, which I loved, but I’m not the only reader, so I would love to see those strengths shine in the paper. I would take the approach you took in responding to these questions in the actual paper content. For example, the statement of “The crowd/people will not be able to apply ML to a new area of their own interest if they can only contribute features to pre-existing problems in pre-existing areas.” is a much stronger statement than “Perhaps…?” (line 39-40) style of writing. Or “research gap” is a great phrase to use to position this paper. This point (of non-experts needing to define their own tasks in new areas to take part in ML) is so critical to this paper. I would strongly encourage stating this as a problem directly, rather than as a question.

Nit: The term “ML problem” and “ML prediction task” seem to be used interchangeably, and I would encourage the authors to provide clarity on these two or choose one. Are they identical or is there a nuanced distinction here that the readers need to understand? “ML prediction task” sounds to be closer to a function that maps an input (feature vector) to an output (label) (e.g., given TF-IDF counts from an email, what’s the probability that it’s related to travel or not?) whereas “ML problem” might be closer to describing a domain area that the non-experts might be familiar with (e.g., favorite travel destinations for hikers). Or perhaps, there is no distinction such that one term should be used throughout the paper.

IRB information seems to be duplicated. (line 282)

Line 503: Should it be “to power “pre-defined” machine learning tasks”? Powering ML methods seem more complex (computational/mathematical) rather than providing labels and features.

Experimental design

Again, thank you for providing clarity in the organization of the research questions and sections as well as thoughtful responses to questions/comments raised. I have a few high-level comments that hopefully can help improve the paper.

I really appreciate the extra description on the definition of learnability of a task. I have to be very honest that this is perhaps the hardest part of the paper to understand still. Again, the experiment design and results are valid. The issue is in understanding the transition from learnability to class balance to perceived/estimated class balance by crowdworkers. Based on the stated definition of learnability ("the ability for a predictive model trained on data collected for the prediction task to accurately generalize to unseen data") and one potential measure of learnability (“balance of class labels”), learnability is a property of the task AND the dataset (potential or train) rather than a property of the task by itself. Does your definition of a prediction task also include (potential) training data? The authors discuss this distinction in lines 391 (“this does not mean the questions are not meaningful, only that the data collected does not lead to useful predictive models”). So, what’s learnable? The task? The task + a train dataset? The task + a true distribution?

Another source of confusion is that class balance is one of many factors that influence learnability. It would be great to provide examples of other factors that influence learnability of a task (maybe...choice of models, features, label quality/consistency, VC dimensions from PAC theory?). The choice of “class balance” as one measure of learnability is understandable because it needs to be something that the crowdworkers can evaluate. But class balance seems to be an oversimplification of learnability. Class balance in the dataset is an important aspect, but moderate imbalance in the dataset is certainly learnable, especially if the prediction task is highly niche (the motivation for this paper is that we want to enable non-experts with their potentially niche problems to be able to ideate prediction tasks). The term “learnability” might be the red herring here. If it was just labeled as “estimated label balance” (such that the model can come up with a set of possible hyperplanes), it would be so much more straightforward. OR, it might be more helpful to position this in an AutoML scenario, where class balance is possibly the only thing that the non-expert should control (selection of model, feature engineering, ML theory is all out of scope for them?). So, instead, you can position "estimated/perceived class balance" as the only "controllable" factor for learnability by non-experts.

Validity of the findings

I was very happy with the findings in the first round, and the additional restructuring and clarification made the paper even stronger.

Method, analysis, and results for RQ1 seem to be sprinkled throughout the paper. What percentage of the solicited prediction tasks were meaningful (learnable AND important)? The analysis seems to decouple learnability (model performance, class balance) separately from importance (subjective rating), as also mentioned in discussion. Should RQ1 be rephrased to “Can individuals … design important or learnable prediction tasks”? Is there a value in defining “meaningfulness” as learnability plus importance in methods if meaningfulness (together) is only going to be mentioned in discussion?

Additional comments

Thank you for your detailed responses and revisions!

·

Basic reporting

The article presents a study with five sub-studies on the question if laypeople are able to design machine learning tasks, what task they design, if they are learnable, and how the quality of the tasks gathered can be increased.
In general, the article is written outstandingly well, the overall research methodology is sound, and the question addressed is quite relevant and well framed and argued in the article. Actually, this is one of the best articles I reviewed in the past months and in my opinion, the article can be published with only minor corrections (technically this would be a minor revision, but I do not need to look at changes). I hope the comments below help the author to make this article a bit sounder and more legible.

On the highest level, my two key suggestions are: First, a clearer presentation of the applied methodology (and the findings) in writing and also in form of illustrations (one bigger for an overview of the sub-studies and one for each study). Second, if possible, to make the article a bit more concise. I found the general approach and the breadth of this work really outstanding. However, the current form of the manuscript is in some parts a bit repetitive (due to the structure of presenting all methods first and then all results instead of an interleaved approach) and has a few —from my perspective—unnecessary “side stages”, such as the discussion about the exact procedure to rank the tasks (section 3.2.1, a reference would be enough, I would assume).

Experimental design

The experimental design of all sub-studies and the overall story of all studies combined is sound. My key reservations stem from the sometimes limited sample size(s) and—more importantly—the fact that it builds on MTurk (see for example https://link.springer.com/article/10.3758/s13428-019-01273-7 and other studies for an explanation of the limitations) for recruiting participants.

My suggestion would be to make the sample sizes and the number of targets generated/evaluated much clearer within the first sentences of the respective sections. Currently, all information is provided but not necessarily at the same spot or in the same structure (this would help readers to get this information much quicker).

Section 4.3 is interesting and relevant, but I had troubles following your approach. I would assume that a model is sufficient/learnable/… if it makes a significant better suggestion than a random choice. Your explanations sound more complicated (and with my background in social science, I had troubles following them completely ;) maybe you check if this section is clear enough.

Validity of the findings

Based on the sound approach, the findings appear to be quite sound. The only bigger reservation I have is the use of MTurk instead of other samples, but this is clearly mentioned in the article (and the implications might be discussed a bit more).
There are some smaller issues than can easily be fixed (see below).

Additional comments

The title is a bit too broad and should more clearly reflect the nature of the tasks collected. I also have some troubles with the breadths of the prefix “DemocratizingAI”, which is excellent “clickbait”, but does not fully reflect the work that was done in this article. Hence, I would advise the author to change the title, which would in turn also increase the “googleability” of this article. Maybe something like crowdsourcing prediction tasks or so…
Also, the study builds on MTurk cloud workers and thus the term “Democratizing” is especially troubling as these people are the (hidden) backbone of numerous research papers, training sets, and implementations, yet they do not participate sufficiently in the design of the tasks, usually do not get any credits etc., and do not participate from the created added value. Hence, is it really “democratizing” if we, from richer countries, exploit people in poorer countries for generating prediction tasks for fractions of a coffee?

The abstract is a bit vague and does not properly summarize the whole article. I would suggest to a) add the multi-step approach, the respective sample/trial sizes and b) shorten it a bit, as- for example- the last sentence does not really help researchers to quickly screen thousands of articles for relevance (it is currently rather an interesting teaser than an abstract).

The type of prediction tasks collected should be introduced much earlier in the manuscript. I read the article from top-to-bottom and, as the title was not clear about it, had quite some troubles to imagine what prediction tasks were to be crowdsourced. It is presented right in the article, but–at least for my preferences—a bit too late which makes it unnecessary difficult to read the article.

Method:

Most of the method(s) is argued and presented extremely well. Yet, I personally do not like the current structure of the method / result section that presents the five method sections sequentially and then the five results section. Instead, I would suggest presenting the five studies sequentially, as it would be easier to follow. However, I leave this open to the author or the journal’s requirements.


All information needed to understand the method and the results are provided, yet I sometimes had troubles finding them quickly. Thus, I suggest that for each of the individual method sections, the number of participants and the number of collected questions/items/data points should be clearly stated.
The overall research methodology and the individual studies should be illustrated using a diagram. This might, for example also include the sample size, parameters measured, number of items gathered etc. I firmly believe that this would increase the overall legibility of this article.

I find the term “workers” I bit troubling and would suggest using the term participants instead.


Minor tidbits


ll. 78. I don’t think that the term “wisdom of the crowd” is appropriate here, as you are rather doing crowdsourcing (the nuances are a bit different).

Figure 1: In the caption the part for the first figure is missing.
Figure 3: I would suggest adding SE or Ci-95% error bars to the diagram. Also, are factual and health/wellness questions distinct categories?

ll. 348. Suggestion: “To rank the /expected/ learnability…”. Also, I think that this whole paragraph would appreciate some shortening.

ll. 398: Make the results section brief but to not be as explicit about it (the shortness is good, but it reads as if you were unmotivated to write more… just remove the “in brief…” :).

Table 4: Please define a level of significance and stick to it. Usually one would use <.001, <.05, and (if really necessary) <.1. The reporting here is a bit chaotic, as some p’s are reported as p=.12 or p>.1.

ll. 410: Why chi^2 and not a t-test/anova? A factor (experimental condition) with two/levels and one ordinal or even metric variable would speak for an anova, which would also the interpretation and presentation of the results easier.

ll. 414: I see this interpretation of apparent but not significant differences frequently, but it makes no sense. Either there is a difference or there isn’t. Here, you argue that the difference is not significant, yet you discuss the difference in detail. Some reviewers would expect that non-significant differences are not reported at all. I would rather suggest to phrase this more neutral. Group A has x points, group B y points. Yet, the difference is not sig. (p=xxx).

Section 3.2, ll 195/196. A rather philosophical point: I don’t agree with the general definition of a “meaningful” task. I do see, however, that your definition is very useful within the scope of your study. I would suggest clarifying that this is just a working definition for the porpose of this article and that meaningfulness, in general, might be more than ML-compatibility.

---

## Round 0.3 · accepted · Accept

Thanks for submitting your minor revisions. Based on my own read of the article and that of one of the original reviewers, the paper can now be accepted for publication.

The only outstanding task is to proofread the paper to make sure that it is consistent after the minor revisions have been applied. See comment from reviewer #1 pointing out a minor issue where "these results" refers to results presented two paragraphs earlier.

Reviewer 1 ·

Basic reporting

Thanks for the new changes that address the comments. Please make sure to read through to ensure smooth transitions between paragraphs, especially around the newly added parts.

Just a small suggestion. Sometimes it helps to style these constructs that you measure (e.g., perceived learnability, perceived importance) with italics to make them stand out as something that you had defined as a construct earlier.

Experimental design

No additional comments

Validity of the findings

Just two small comments for Discussion section.

Line 460-466: I think it's a bit of overclaiming to state that non-experts can estimate learnability for classification (all of them?). They can estimate learnability for the classification tasks in this paper, but that's still a small set of all classification tasks. Generalizing beyond what's done in the paper should be called out as future works, with potential avenues or dimensions that other researchers should be looking at. Reiterating the study findings is great. But generalizing beyond it is the problem here. In that regard, please be careful in separating out what you've found in your work and what can be generalized beyond that.

Line 467: With the insertion of the learnability paragraph, this doesn't flow anymore. "these results" you refer to appear in 2 paragraphs ago. Could you make sure your stories are consistent here?

Additional comments

I'm happy with the latest changes around learnability and clarifications that were addressed based on Reviewer 3's comments.